# Compact Imaging Systems Based on Annular Harmonic Lenses

**DOI:** 10.3390/s20143914

**Published:** 2020-07-14

**Authors:** Roman Skidanov, Yury Strelkov, Sergey Volotovsky, Veronika Blank, Sofiya Ganchevskaya, Vladimir Podlipnov, Nikolay Ivliev, Nikolay Kazanskiy

**Affiliations:** 1Image Processing Systems Institute of RAS—Branch of the Federal Scientific Research Centre Crystallography and Photonics of Russian Academy of Sciences, 151 Molodogvardeyskaya st., 443001 Samara, Russia; strelkovUS163@gmail.com (Y.S.); sv@smr.ru (S.V.); veronika_b@ipsiras.ru (V.B.); podlipnovvv@ya.ru (V.P.); ivlievn@gmail.com (N.I.); kazanskiy@ssau.ru (N.K.); 2Department of Technical Cybernetics, Samara National Research University, 34 Moskovskoe shosse, 443086 Samara, Russia; sofi@ipsiras.ru

**Keywords:** harmonic lens, imaging system, ultracompact long focus objective, annular aperture

## Abstract

In this study, a configuration of a compact imaging objective based on a reflecting annular harmonic lens was proposed. Light propagation through the proposed optical system was comprehensively modeled using a dedicated special program and the ZEMAX software, with the latter used to derive the point spread function (PSF). Several relationships were used to describe the connection between key parameters of the objective, including its focal length, field of view, and thickness. We demonstrated that it was possible to design a compact imaging objective whose overall length could be one to two orders of magnitude smaller than its focal length. Using direct laser writing, a reflecting annular harmonic lens was fabricated and used in the proposed objective scheme. The performance of the objective was experimentally studied by imaging a light source and a test pattern. The performance of the compact imaging objective based on a reflecting annular harmonic lens was verified in principle. A PSF value of approximately 16 microns was experimentally obtained, for a lens with a diameter of 25 mm with a focal length of 100 mm.

## 1. Introduction

The minimization of imaging systems has been a recent trend, with increasingly compact devices emerging that allow high-quality imaging. However, in one class of imaging systems, miniaturization is restricted by laws of optics, such as in remote surveillance systems that use long focus objectives. For such system requirements, in order for the acquired imagery to be of a specified scale, such imaging devices cannot be essentially miniaturized. For instance, in the Nikon AF-S 600mm f/4E FL ED VR Nikkor camera, the objective is more than 400 mm long and weighs almost 4 kg. The mentioned objective is a product of a lengthy evolution, during the course of which its mass and size have reached a minimization limit, making further reduction in size based on classical optical components impossible. This leaves the question, “What are we expected to do if the objective size needs to be comparable with or slightly larger than the size of a sensing array, for instance, if we plan to send a space-borne computer vision system on a mission to Alpha Centauri [1] or mount it on a microdrone?”

When seeking to minimize imaging systems, it makes sense to use diffractive optics capabilities. Imaging systems that depend on the use of diffractive lenses are essentially smaller in mass and more compact as compared with imaging systems based on refractive lenses. Wider use of diffractive lenses is prevented due their high chromatism [2], which are partially compensated for by using harmonic lenses [3,4,5,6,7,8,9,10]. Chromatic aberrations can partly be compensated for by using post-capture image processing [11,12,13]. The use of harmonic-lens-aided imaging systems enables the objective mass to be reduced because the harmonic lens thickness equals that of the microrelief substrate [4]. Harmonic lenses can be used both in the visible and in the mid-IR spectral ranges [14,15,16,17,18,19] but are unable to help reduce, for example, the geometric size of a telescope objective, thus, requiring a fundamentally new technical solution. The size of an imaging objective can be significantly reduced by developing an approach based on a two-mirror telescope system [20].

Basic schemes of two-mirror telescopes were proposed by Cassegrain and Gregory in the 17th century. In their classical versions, the primary mirror represented a paraboloid of rotation. An image of an on-axis point at infinity constructed by the major mirror at its focus (F) was transferred to point F1 where the secondary mirror was placed. In the Cassegrain scheme, the secondary mirror was located between the primary mirror and its focus was to preserve zero spherical aberration, with the secondary mirror shaped as a convex hyperboloid with its focus at point F. In the Gregory scheme, the secondary mirror was located behind the primary focus F and the mirror was shaped as a concave ellipsoid with its foci at points F and F1, and therefore the on-axis image remained sharp. Because the focal length fits twice into the telescope overall size, the Cassegrain scheme was more compact. In previous studies [21,22], an idea was proposed for a new imaging system which was composed of two mirrors and a long focus reflecting annular harmonic lens that has a longitudinal dimension of just several millimeters.

In this paper, as a further development of the said design, we have proposed an imaging system composed of two mirrors and a long focus reflecting annular harmonic lens that has a longitudinal size of several millimeters. We performed geometric optics-based modeling in the HARMLENS special program, as well as in the ZEMAX program. We also conducted several experiments with the laboratory layout of the proposed design. 

## 2. Design of a Two-Sided Imaging Objective on a Plane Plate 

Here, we analyze a telescope system (with an infinite object segment) as a development of the Cassegrain design. Let the secondary mirror be plane, and the primary mirror also be plane but with a reflecting microrelief along the edge, realized as a harmonic lens, that is, the objective consists of a reflecting annular harmonic lens and two mirrors, with an output opening at the center of the bottom mirror to let the focused light out (Figure 1a). Leaving aside multiple reflections, the equivalent system is similar to the Cassegrain design (Figure 1b). It stands to reason that in this system, the principal planes and the input pupil are found in the plane of the harmonic lens. Similar to the Cassegrain design, the output pupil is found behind the secondary mirror and imaginary. 

In calculations, the lens is assumed to be a harmonic parabolic lens. The surface shape of a transmission harmonic lens [3] is given by:(1)$$h\left(r,\lambda_{0}\right)=\frac{1}{n-1}MOD_{m\lambda_{0}}\left(\frac{k}{2f}r^{2}\right),\;m=1,2,3,4\ldots$$
where *λ*_0_ is the central design wavelength, m is the number of harmonics, and the $MOD_{m\lambda_{0}}$ operation denotes that the maximum ray path difference in this lens is *mλ*_0_.

For a reflecting lens, the formula is modified to:(2)$$h\left(r,\lambda_{0}\right)=\frac{1}{2n}MOD_{m\lambda_{0}}\left(\frac{k}{2f}r^{2}\right),\;m=1,2,3,4\ldots$$

Therefore, although the microrelief height is relatively small, the number of harmonics of the reflecting lens is sufficiently large to ensure that the chromatic aberrations due to diffraction of light are not higher than classic chromatic aberrations due to dispersion in the lens material.

The modeling is conducted for a 50 mm parabolic lens with a 100 mm focal length. The microrelief height of 10 µm was chosen based on the capabilities of the technique of direct laser writing in a photoresist [23,24,25,26], which is used for synthesizing harmonic lenses. For the central wavelength of λ_0_ = 0.5 µm, the lens will have m = 30. A profile of the lens microrelief for the aforesaid parameters is demonstrated in Figure 1c. The cross-section was obtained by direct calculation using Equation (2).

Because of multiple consecutive reflections, this system ”squeezes” the telescope objective almost to the size of a plate of thickness 2–10 mm, making it possible to design imaging systems with a focal length of several hundred millimeters. The length of the proper objective, when considering the distance to a photosensor array, is an adjustable parameter that can amount to just 1–10% of the focal length.

There can be a situation where the plate thickness is less than 1 mm while the working segment length is zero. In this case, the objective is transformed into a thin plate that can be located directly on the surface of a photosensor array.

For the aforesaid parameters, the angle α at which the outermost lens zone reflects the beam is ~14° (α=arctg(D/2f)). 

Let us analyze the key relationships that describe this system. There are several constraints that are imposed on the parameters of the proposed system. As a starting point, consider constraints that exist when the system deals with a point source on the optical axis. The first constraint is that the central opening radius should not be less than the focal length multiplied by the tangent of the maximum angle of the field of view (FOV). If the opening becomes smaller than that, it turns into an aperture diaphragm. The minimal system length is determined by the plate thickness s and a quantity df, which, in turn, is connected with the minimal size of the central opening:(3)$$df=f\frac{d}{D}$$
where D is the lens’ diameter and d is the central opening diameter. 

The maximum width of the annular aperture, Δ*r*, is also restricted by the relation: (4)$$\Delta r=s\frac{D}{2f}$$
where s is the plate thickness. 

From Equation (4), the aperture ring width is seen to be proportional to the plate thickness s. With decreasing thickness, the light-gathering power of the telescope tends to zero; however, in several cases it still remains sufficient for practical uses.

Due to multiple reflections within the system, the reflectance needs to be high. The number of reflections in the system is defined by: (5)$$N=\left[\frac{D-d}{2D}\frac{f}{s}\right]$$
where [] denotes the integer part of the number. Thus, defining the reflectance as β, the system energy efficiency η is given by:(6)$$\eta =\beta^{N}=\beta^{\frac{D-d}{2D}\frac{f}{s}}$$

The thinner the plate, the larger the number of reflections, and the smaller the amount of light making it through the optical system. For estimation purposes, using s = 10 mm, D = 50 mm, d = 5 mm, f = 100 mm, and β = 0.95, we found that η = 0.63, with the overall objective length being as small as 20 mm. The light intensity in this system can be increased by using several ring apertures, as well as more advanced mirrors. The aperture of this system increases when creating a relatively short-focus lens of the same design.

When operating with an off-axis point object, still more constraints are imposed on the system (Figure 2).

Let us analyze the propagation of a tilted parallel beam through the system in Figure 2. When the beam is tilted to the optical axis, the two factors that affect the imaging quality are as follows: On the left, a portion of rays passes by the reflecting lens, whereas on the right, if the tilt angle is larger than α/2, light does not enter the optical system. Although the first factor is easy to eliminate by making the microrelief area wider, the second is irremovable, with the field of view being limited by the angle of α/2. For the aforesaid parameters, this angle is approximately 7°. Note, however, that these constraints are linked not with the original system in Figure 1a, but with its equivalent in Figure 1b. In the original system, there is a harder constraint on the FOV associated with a different number of rays that enter the system from different sides (Figure 3).

Thus, the major constraint on the FOV of the system under study follows from the necessity to ensure an equal number of reflections for the rays entering the objective from different sides of the objective.

If the angle of reflection on the edge is α and the beam is tilted by β, then, the light travels after reflecting at an angle of α + β on the left and at an angle of α – β on the right (Figure 3).

Then, the number of reflections on the left is:(7)$$N_{1}=\left[\frac{D-d}{4s\cdot tg\left(\alpha +\beta \right)}\right]$$
where [] denotes the integer part of the number, with the number of reflections on the right being:(8)$$N_{2}=\left[\frac{D-d}{4s\cdot tg\left(\alpha -\beta \right)}\right]$$

Substituting the above-specified parameters of the imaging system into Equations (7) and (8) yields the equality of the number of reflections for angles smaller than 1°. With a small focal length as compared with the diameter, this design allows for significantly larger angles, therefore, at f = 20 mm, while maintaining all other parameters, the angle of view would be approximately 5°. However, the technology used for the production of harmonic lenses currently does not allow the production of lenses with such parameters.

## 3. Numerical Ray Tracing for the Proposed Optical System 

The proposed imaging system was optically modeled using the in-house software HARMLENS developed for modeling harmonic lenses, and the results were verified using the commercial software ZEMAX (Zemax, LLC, Kirkland, Washington). The HARMLENS program considers diffraction effects by calculating the phase function of a harmonic lens for a given wavelength. According to this phase function, the angle of deflection of the beam at a given point in the harmonic lens for a given wavelength is determined.

Let us analyze a situation where the harmonic lens and the secondary mirror are located on the surface of different silica plates. Figure 4 shows the ray tracing (side view) when light is focused directly on the surface of the first mirror assuming that D = 50 mm and f = 100 mm, with the parameters d and s being varied to illustrate the system performance under different conditions. 

Figure 4 shows the proposed configuration which allows an imaging system design with a zero-length working segment. Decreasing the distance between the primary and secondary mirrors leads to decreasing the width of the input annular apertures and decreasing the output openings. In the examples above, the ratio of the focal length and the annular aperture diameter are fitted to make the output opening diameter approximately equal to the distance between the primary and secondary mirrors.

By varying the inter-mirror distance and the output opening diameter, it becomes possible both to increase the working segment length (Figure 5a,b) and to shift the imaging plane toward the secondary mirror plane (Figure 5c). Figure 5 shows an imaging system with D = 50 mm and f = 250 mm. In Figure 5a, the mirrors are 5 mm apart and the working segment is 100 mm long. In Figure 5b, the mirrors are still 5 mm apart, whereas the working segment is increased to 20 mm due to the expanded output aperture. In Figure 5c, the mirrors are 4.5 mm apart and the beam is focused in the secondary mirror plane. 

By focusing light on the secondary mirror plane, it becomes possible to place the photosensor array in front of the optical system, which could be more convenient in some situations. The system FOV is limited by a different number of reflections of the light field differently tilted at different aperture edges. In Figure 6, the situation is illustrated by the example of a lens with D = 50 mm, f = 250 mm, and an inter-mirror distance of 10 mm, illuminated by a plane wave at an angle of 0.12°.

At first glance, the light is focused on the first mirror plane; however, a more detailed analysis reveals that the light that enters the system from the left, after experiencing one extra reflection, is focused on the second mirror surface, arriving at the first mirror essentially defocused. While hardly deteriorating the image quality, this leads to an effect of generating, in addition to the principal FOV, a secondary FOV where a less bright image forms. To verify the validity of modeling, a lens with d = 50 mm and f = 100 mm was also numerically simulated using ZEMAX (Zemax, LLC, Kirkland, Washington) software. 

Figure 7 shows ray tracing for an on-axis beam of rays passing through the system in two-dimensional (Figure 7a) and three-dimensional (Figure 7b) cases.

The system was observed to have a focal point, and therefore represents an imaging system that is at least valid for on-axis points (Figure 7). Next, the point spread function (PSF) was constructed. As ZEMAX was unable to automatically determine the focal plane position for the imaging system under analysis, this was achieved by scanning a small on-axis segment with a recording surface. The intensity distribution in the system was given by an intensity ring at any plane except for the focal plane (Figure 8a). The intensity distribution derived via scanning in the focal plane was found to be fairly compact (Figure 8b), representing the PSF of an imaging system of the classical form. Figure 8c shows the radial profile of the PSF, from which its width can be derived.

The PSF width at half maximum is approximately 4 µm, which is a fairly good value as compared with the PSF width of refractive objectives. Thus, we inferred that the system was suitable for imaging purposes. Due to the annular input aperture, the diffraction PSF of the system is different from the PSF of classical imaging systems with circular apertures [21]. As a rule, the diffraction PSF is defined as a ratio of the first-order Bessel function to its argument $\frac{J_{1}\left(kar\right)}{kar}$, where *k* is the wavenumber and *a* is the input aperture radius. For an annular input aperture, the diffraction PSF is defined by a zero-order Bessel function, $J_{0}\left(kar\right)$. The width of the central peak of the *J_0_* function is slightly narrower than that of the *J_1_* function; the height of the secondary rings is, conversely, higher than that of the *J_0_* function. Thus, in a diffraction-limited variant, the annular system resolves point objects better (e.g., stars) but has a worse modulation transfer function (MTF).

Let us analyze the performance of this objective when imaging off-axis points. For this purpose, we assume a parallel input beam of rays that creates an angle of 0.7° with the optical axis, remaining in the objective FOV (for D = 50 mm and f = 100 mm). Figure 9 presents ray paths in two-dimensional (Figure 9a) and three-dimensional (Figure 9.b) cases, and the corresponding PSF. 

The PSF was observed to be wider as compared with the imaging an on-axis point (Figure 9c). The PSF width at half maximum is approximately 10 µm, which means an essential growth of aberrations. There was an aberration of the coma, based on the form of the PSF. Even at such a PSF, the objective has a resolution of approximately 50 l/mm, which is acceptable for practical uses, that is, this objective is an imaging system and can be used in combination with modern photosensor arrays, whereas the use of post-capture image processing [27,28,29,30] enables the acquisition of an image comparable in quality with that of refractive objectives. Thus, the proposed imaging system could have practical uses in areas where compactness is of greater significance than the lens aperture. 

## 4. An Optical Experiment on Image Acquisition Using the Proposed Imaging Scheme 

An optical experiment was conducted using a prototype imaging system with an annular aperture. The annular aperture was fabricated by direct laser writing in a photoresist and, subsequent, magnetron deposition of an aluminum layer. The focal length of the annular lens was 100 mm, the diameter of the prototype lens was 25 mm due to fabrication process limitations, and the width of the microrelief ring was 0.75 mm. The design wavelength of the lens was 650 nm. An external view of the fabricated annular lens is shown in Figure 10a, the lens profile over the entire microrelief width is shown in Figure 10c, and the overall system appearance in a plastic housing is shown in Figure 10b (manufacturing error 20 μm). With the secondary mirror being placed 10 mm from the primary mirror, the rays should experience five reflections in the system.

The experiment was conducted using a simple optical setup (shown in Figure 11) which included a ToupCam UCMOS03100KPA camera (Hangzhou ToupTek Photonics Co., Ltd., Zhejiang) with 5.5 µm pixels. The focusing plane was in the harmonic lens plane but because the object under imaging was at a finite distance of 800 mm, which was comparable with the focal length, the imaging plane was found to be 12 mm outside the objective. Thus, with the photosensor array located inside the camera housing, the objective was placed right up to the housing to obtain a sharp image. If an infinitely remote object was to be imaged, the objective would have to be placed directly near the photosensor array. We experimented with a harmonic lens with a minimum working distance and built a lens that produced an image of an infinitely distant object on its back surface. However, in this case, this lens had to stand directly on the matrix, which was difficult to do, since the matrix already had an IR filter. Thus, the close location of the object served to slightly increase the working segment.

The objects under imaging included a test pattern (Figure 12), two small-area light-emitting diode (LED) sources, with one LED composed of red, green, and blue triplets (RGB-LED), and the other LED composed of 16 circularly arranged white sources, as well as an incandescent lamp.

Because the system has a small FOV, only a small fragment of the test pattern (the marked area in Figure 12) was recorded with the photosensor (Figure 13a,b).

As expected from optical modeling results, two different regions in the FOV are shown in Figure 13a, i.e., the central region of higher intensity and the peripheral region of lower intensity, with the latter being formed by the light coming from a portion of the aperture. The second region had an asymmetry, likely because the mirrors were slightly mutually tilted. Post-capture processing that involved background noise removal partly improved the image quality, but the lower-intensity FOV region became hardly discernible. The image contrast was estimated at 0.05 to 0.09 based on the cross-section (Figure 13b). With the low light-gathering power of the objective hampering the imaging system characterization based on the image in Figure 13, the PSF width was evaluated using images of LEDs and an incandescent lamp (Figure 14).

Figure 14 shows that the images from high-intensity light sources are fairly high contrast. For instance, at its narrowest, the coil image is just two pixels wide (11 µm). However, all the shortcomings of harmonic lenses are inherent in this system. The analysis of Figure 14c suggests that the upper row of red LED images has the best quality because the design wavelength of the lens is 650 nm. The blue LED images are hardly discernible, not only due to defocusing but also due to the reduced diffraction efficiency. In Figure 14b, images of some white LEDs are found in a FOV region where light does not come from the entire aperture. In this region, emitting surfaces can be observed due to the lower brightness, but aberrations are high. To evaluate the light source size, Figure 14b depicts a combined image of simultaneously emitting LEDs and an incandescent lamp, with the coil seen at the image center. An estimate of the PSF width at half maximum was obtained on the edge of the light source image. 

An estimate of the polychromatic PSF in the combined image of the incandescent lamp and the white LED has a width at half-maximum of 2–3 pixels (11–16.5 µm). For individual spectral ranges, the PSF width at half maximum has the following different values: red LEDS, 2 pixels (11 µm); green LEDs, three pixels (16.5 µm); and blue LEDs, 4 to 5 pixels (22–27.5 µm). The measurements were conducted for an object placed near the optical axis. Although the width of the on-axis PSF is much larger than the optically modeled value, but considering harmonic lens properties [2], this result can be expected. In terms of practical applications, we have proposed a totally workable imaging system design, which needs to be further improved and developed. Even at the current stage, the proposed configuration has enormous potential for further development as a basis for compact objectives. 

## 5. Discussion

We demonstrated a feasible design of a long focus objective whose overall size and working segment are one to two orders of magnitude smaller than its focal length. There are no physical constraints on the reduction of the objective length. It is possible to design an objective with a 1 mm distance between the primary and secondary mirrors and a zero-length working segment, and thus obtain a working imaging system. Therefore, the imaging system can be transformed into a planar sensor in which the optical part and the photosensor look like two closely placed plates. As the photosensor array is normally protected by a silicon oxide film, the protective film can be replaced with the proposed objective design. However, the practical use of the proposed objective concept is limited by a number of factors. First, with a decreasing distance between the primary and secondary mirrors, the FOV of the objective is proportionally reduced. Second, the objective of this type, in principle, has a low light-gathering power, and therefore can practically only be used for the observation of brightly illuminated objects. Third, at the current state of technology for diffractive structure fabrication, the image quality is far from being ideal. The majority of these limitations can be overcome. Thus, the small FOV can be compensated for by using a number of identical systems, each capturing its own FOV fragment. This can be achieved because in the proposed objective design, with decreasing inter-mirror distance, the size of the illuminated region in the photosensor array also decreases, enabling the use of small-size sensor arrays. The image quality can be improved by increasing the microrelief height [6] and reducing fabrication errors. Considering multiple reflections, more stringent requirements are imposed on the accuracy of fabricating reflecting layers of both mirrors. Modern requirements of a λ/20 tolerance on the surface roughness (λ is the incident wavelength) are most likely to be insufficient for the proposed objective. The proposed objective design could potentially offer high spatial resolution because, for the annular input aperture, the diffraction limit enables resolving higher frequencies. For the first time, the idea of the proposed system design was introduced during the course of our team’s interaction with the Milner–Hawking’s project [1]. These compact objectives could also find uses in more practical areas. It would be convenient to use them onboard small flying vehicles for remote surveillance. They could also be useful for the observation of high-brightness objects when the light normally needs to be attenuated with neutral optical filters. In the proposed system, the attenuation occurs in a natural manner due to a narrow aperture and can be varied by varying the inter-mirror distance. Thus, the results of the optical and computational experiments demonstrated the fundamental feasibility of designing an ultracompact objective based on the proposed optical scheme.

## 6. Conclusions

An optical experiment confirmed the imaging properties of the proposed optical system. A compact annular planar objective composed of a harmonic lens and two mirrors on the opposite surfaces of a plate allowed the image acquisition. The theoretically derived on-axis value of the PSF (approximately 4 µm width) is close to the diffraction limit. In an optical experiment, the PSF was estimated at 11–16 µm, which is sufficient for the practical use of the system in computer vision systems for which the overall size is critical, such as microdrones, covert surveillance cameras, pico-satellites, and femto-satellites.

## Figures and Tables

**Figure 1 sensors-20-03914-f001:**
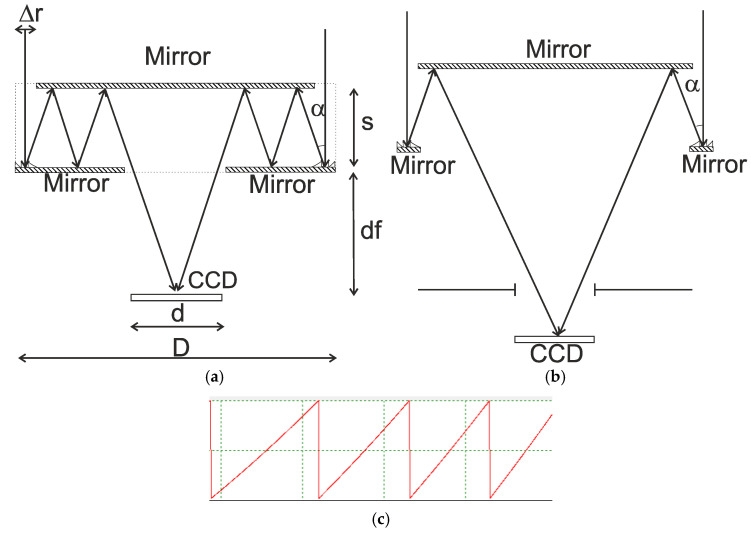
(**a**) A planar annular imaging objective; (**b**) Its equivalent without multiple reflections; and (**c**) The total cross-section of the microrelief of an annular harmonic parabolic lens of height 10 µm. CCD, charge-coupled device; D is the lens diameter; d is the central opening diameter; s is the plate thickness; Δ*r* is width of the annular aperture; and df is working distance.

**Figure 2 sensors-20-03914-f002:**
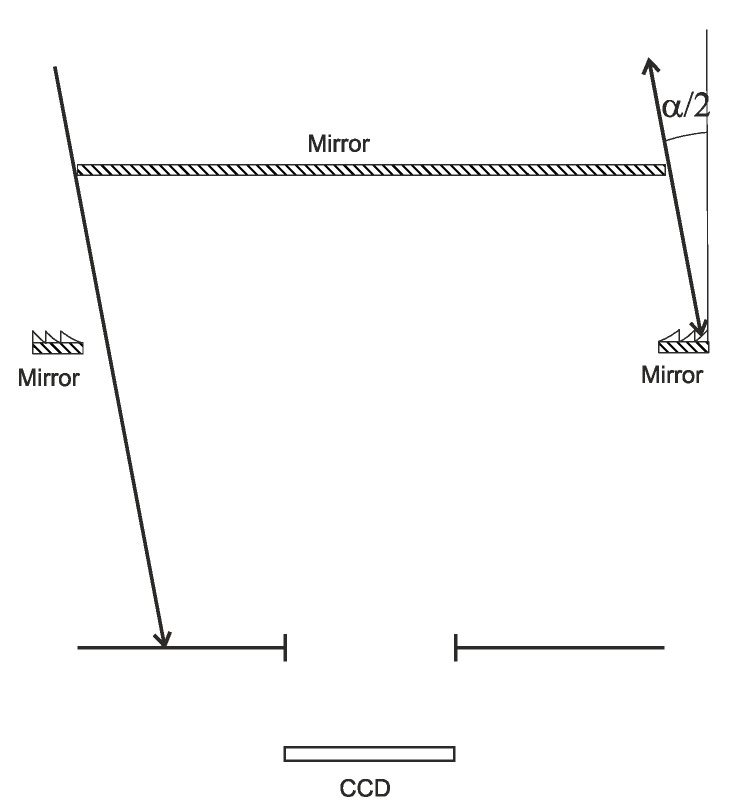
Travel of a light beam that makes an angle of α/2 with the optical axis through the objective.

**Figure 3 sensors-20-03914-f003:**
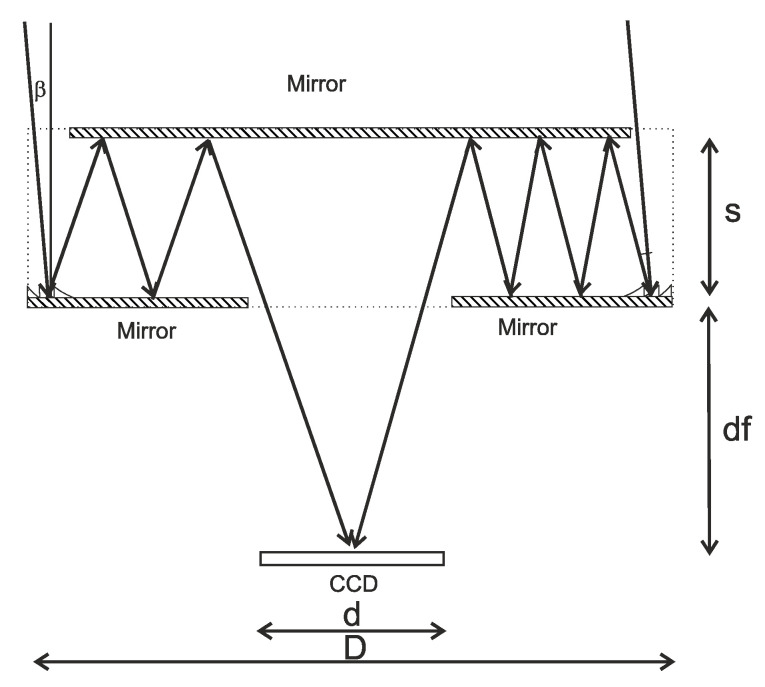
Incidence of a tilted beam with a different number of reflections.

**Figure 4 sensors-20-03914-f004:**
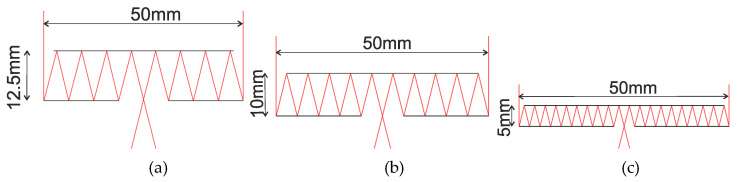
Ray tracing when focusing on the surface of the first mirror in an optical imaging system with an annular aperture D = 50 mm, *f* = 100 mm. (**a**) s = 12.5 mm and d = 12.5 mm, on-axis plane wave; (**b**) s = 10 mm and d = 10 mm, on-axis plane wave; and (**c**) s = 5 mm and d = 5 mm, on-axis plane wave.

**Figure 5 sensors-20-03914-f005:**
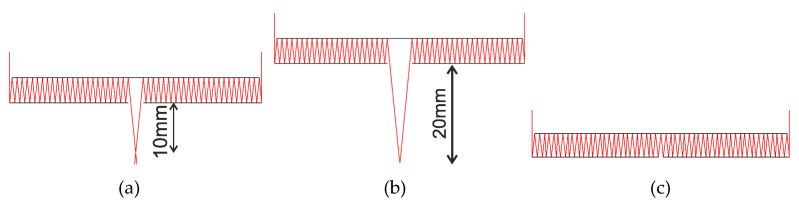
An imaging system with D = 50 mm and f = 250 mm. (**a**) The mirrors are 5 mm apart and the working segment is 10 mm; (**b**) The mirrors are 5 mm apart and the working segment is 20 mm; and (**c**) The mirrors are 4.5 mm apart and focusing is on the secondary mirror plane.

**Figure 6 sensors-20-03914-f006:**
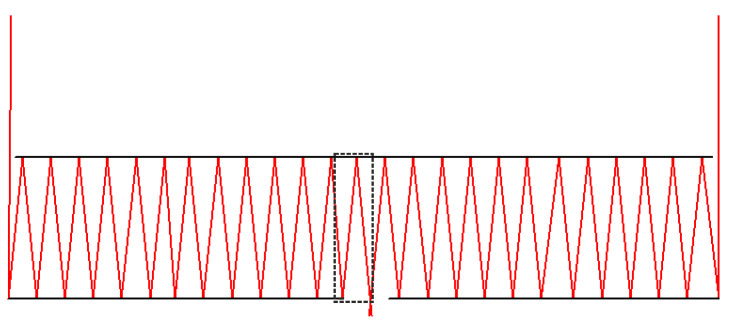
Focusing a tilted light wave that leads to a different number of reflections of the light coming from the right and left. We have highlighted an extra reflection with a dotted line.

**Figure 7 sensors-20-03914-f007:**
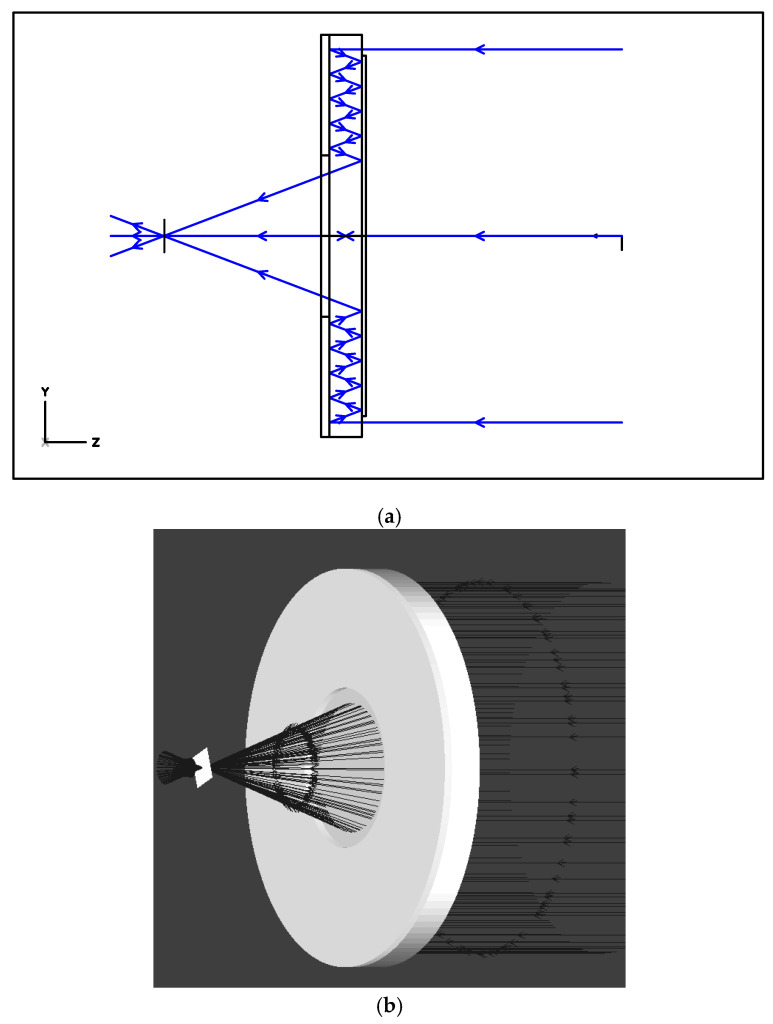
Ray tracing through a planar annular imaging objective. (**a**) Two-dimensional; (**b**) Three-dimensional cases.

**Figure 8 sensors-20-03914-f008:**
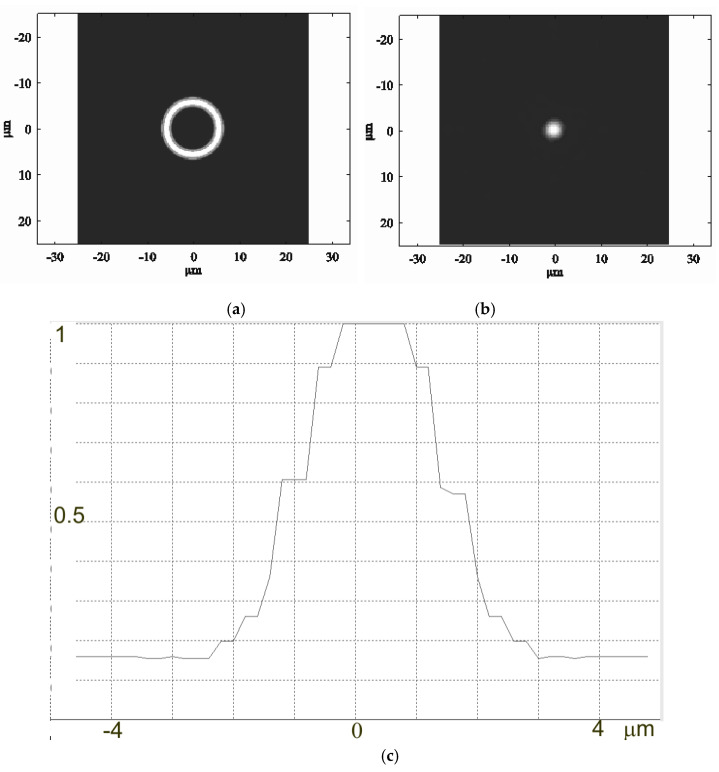
(**a**) A typical form of the intensity distribution on the recording surface outside the focal plane; (**b**) Point spread function (PSF); (**c**) PSF profile.

**Figure 9 sensors-20-03914-f009:**
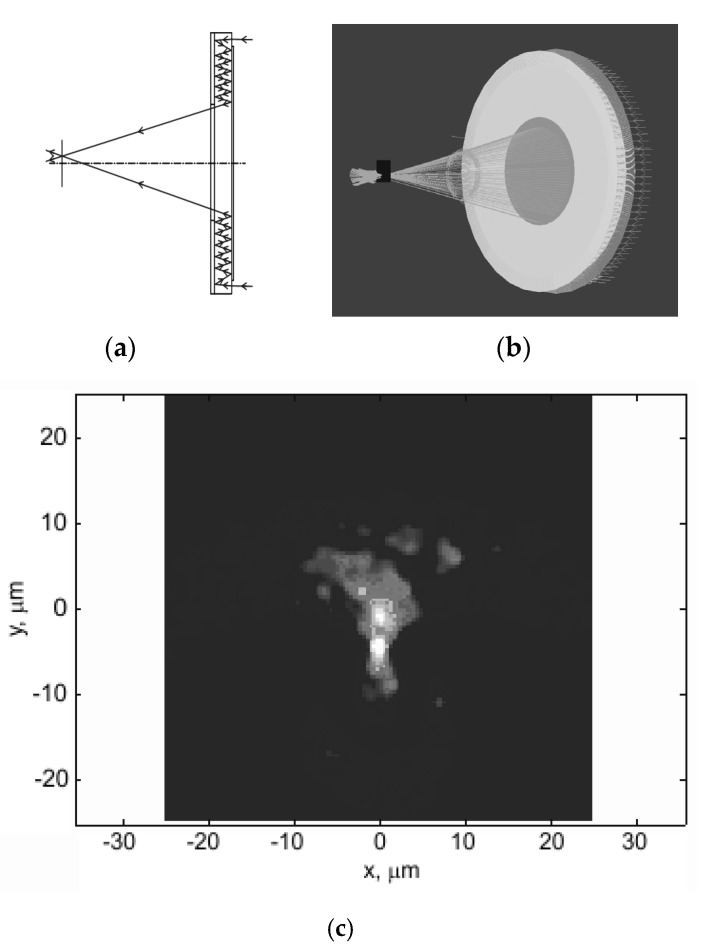
Ray paths in (**a**) two-dimensional and (**b**) three-dimensional cases, and (**c**) PSF.

**Figure 10 sensors-20-03914-f010:**
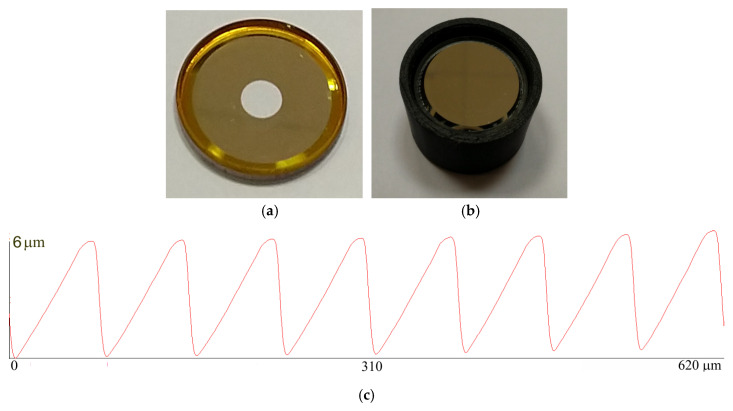
(**a**) External appearance of the fabricated reflecting harmonic annular lens with an annular zone with microrelief seen along the edge; (**b**) The external appearance of the imaging system with annular aperture as seen from the secondary mirror side; (**c**) The microrelief profile.

**Figure 11 sensors-20-03914-f011:**
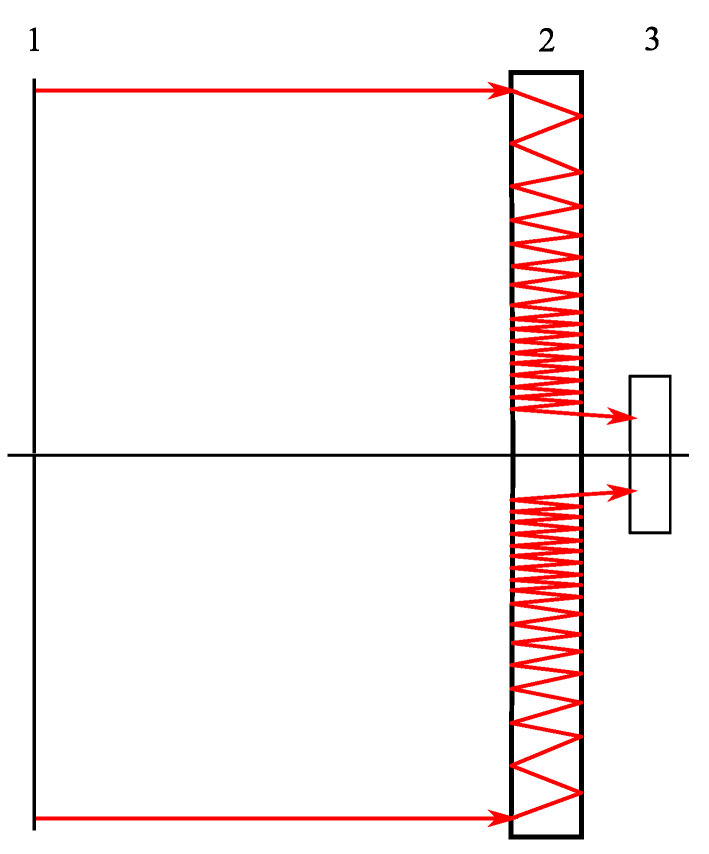
An optical setup for image acquisition using an optical system with annular aperture. 1, a test pattern illuminated by a planar uniform light source; 2, an annular imaging system; and 3, a recording camera ToupCam UCMOS03100KPA.

**Figure 12 sensors-20-03914-f012:**
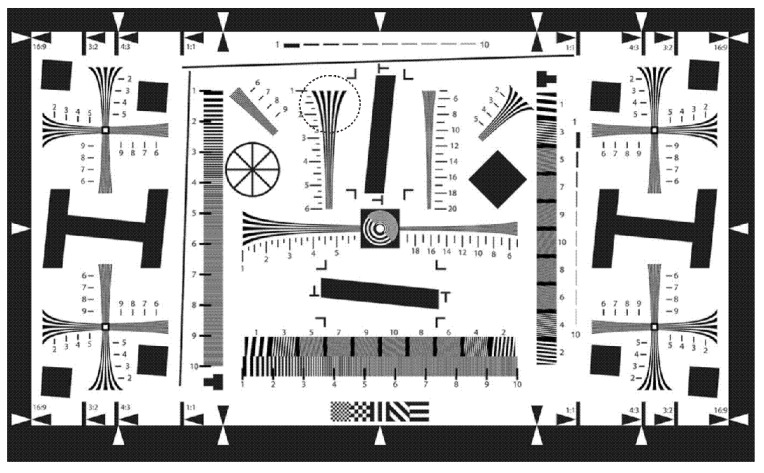
A black-and-white test pattern for testing lenses.

**Figure 13 sensors-20-03914-f013:**
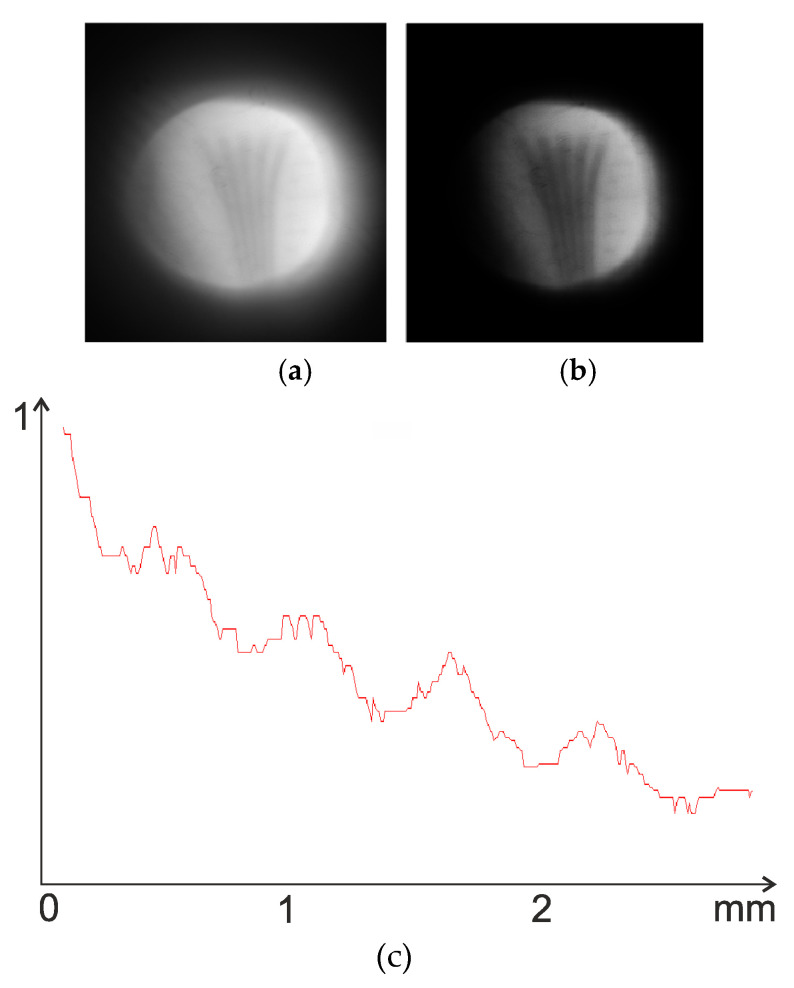
A black and white test pattern for testing lenses obtained in the system with an annular aperture. (**a**) original view; (**b**) After increasing the contrast; (**c**) Cross-section of (**b**).

**Figure 14 sensors-20-03914-f014:**
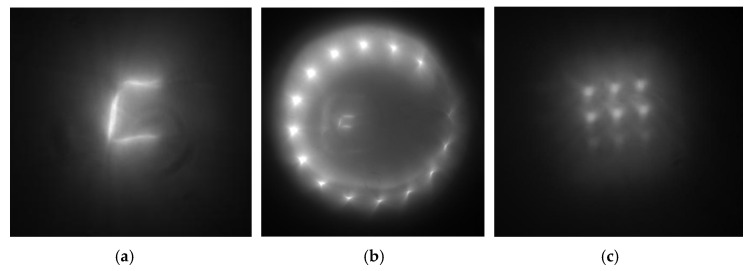
Images of different light sources. (**a**) An incandescent lamp coil; (**b**) White light-emitting diodes (LEDs); (**c**) Color LEDs (upper row, red; middle row, green; and lower row, blue).

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
