# Peer review of "Compact Imaging Systems Based on Annular Harmonic Lenses"

_sensors, 2020, doi:10.3390/s20143914_

Round 1

Reviewer 1 Report

In order to create a new optical system, improvement is also required in the traditional lens, and this paper deals with this new type of lens.

However, I think that there should be additional explanation on the following contents in order to be published in this paper.

1. Both front and back sides are mirror surfaces. In particular, since the front side is a mirror, the light is obscured. In this case, F/# of the optical system may be large, so it is considered that a bright optical system cannot be formed. it is necessary to explain if there is a way to improve this.

2. When using this lens, it is necessary to mention whether it can be applied to optical systems with a large field of view.

Reviewer 2 Report

This paper entitled "Compact imaging systems based on annular harmonic lenses" has originality and after a major revision of its text and figure I believe that this paper can become of great interest for sensors.

The authors are trying to demonstrate that it is possible to design a compact image lens whose total length can be one to two orders of magnitude smaller than its focal length.

However, there are some modifications to be followed:

* There are no references and comments in the introduction to the modeling process described in the abstract and throughout the text.

For example, the Zemax software, with the one used to derive the point spread function (PSF).

* Figures 1, 2 and 3 in these figures, legible information is not visible. For example the angle α or the words Mirror ...

* It is not clear in the text how it was built
the micro-relief profile of the lens shown in Fig. 1c

* Numerical ray-tracing for the proposed optical system

* Optical modeling of the proposed imaging system is conducted using the in-house software
HARMLENS developed for modeling harmonic lenses but would like a more detailed description.

* In figure 7 you cannot understand the figure. It must be increased.

Figures 8c, 9c and 12 must be redone so that the information is visible.

It would be better if section 5 Discussion were incorporated into the paper's own results.

Reviewer 3 Report

A configuration of a compact imaging objective based on a reflecting annular harmonic lens is proposed in this paper. Comprehensive modeling of light propagation through the proposed optical system is conducted using a dedicated special program and the ZEMAX software, with the latter used to derive the point spread function (PSF).

The imaging experiment was carried out, and the results were compared with the standard test image card. In my opinion, the technology proposed in this paper has potential application value in the field of optical imaging.

But the level of the achievements of this paper should be highlighted in the abstract,such as technical indicators achieved.

Reviewer 4 Report

Line 91 - part in brackets should be omitted;

Regarding paragraph 3. (starting from line 159)
I would recommend adding plots showing relations between dimensions of the optical imaging system to supplement info from Figure 4. and 5.

Figure 6. - extra reflection should be extra shown/indicated. 

Lines 200-201 - showing this "interesting effect" would be nice.

Figures 8. and 9 are too small, not possible to read what is written on each axis. 

Why PSF which is shown on 8c is so "step-like"?  

Figure 9c. - PSF shape discussion should be made (which aberrations, cross-section,  change of PSF with system change, etc.).

Regarding paragraph 4.

What ware the tolerances of assembling the system components?

Where are the origins of such high difference theory - practice? 

Is it possible to simulate the object in infinity by using collimator - lines 266-272?

Figure 13. could you shown which part of the test (Figure 12) the system is actually imaging?

Discussion of results - line 288-320 could be more detailed - contrast (while mention) should be calculated. Cross-sections for illustrations would be appreciated.

Round 2

Reviewer 2 Report

The authors improved the presentation of the paper  "Compact imaging systems based on annular harmonic lenses".

In my opinion, the paper can be accepted for publication in Sensors.